# Interactions between Autophagy and DNA Viruses

**DOI:** 10.3390/v11090776

**Published:** 2019-08-23

**Authors:** Hai-chang Yin, Shu-li Shao, Xin-jie Jiang, Peng-yu Xie, Wan-shu Sun, Tian-fei Yu

**Affiliations:** 1College of Life Science and Agriculture Forestry, Qiqihar University, Qiqihar, Heilongjiang 161006, China; 2Heilongjiang Provincial Key Laboratory of Resistance Gene Engineering and Protection of Biodiversity in Cold Areas, Qiqihar, Heilongjiang 161006, China

**Keywords:** autophagy, DNA viruses, interactions, replication, immunity

## Abstract

Autophagy is a catabolic biological process in the body. By targeting exogenous microorganisms and aged intracellular proteins and organelles and sending them to the lysosome for phagocytosis and degradation, autophagy contributes to energy recycling. When cells are stimulated by exogenous pathogenic microorganisms such as viruses, activation or inhibition of autophagy is often triggered. As autophagy has antiviral effects, many viruses may escape and resist the process by encoding viral proteins. At the same time, viruses can also use autophagy to enhance their replication or increase the persistence of latent infections. Here, we give a brief overview of autophagy and DNA viruses and comprehensively review the known interactions between human and animal DNA viruses and autophagy and the role and mechanisms of autophagy in viral DNA replication and DNA virus-induced innate and acquired immunity.

## 1. Introduction to Autophagy

Autophagy is evolutionarily conserved from yeast to humans and maintains cellular homeostasis by eliminating dysfunctional cellular organelles, protein aggregates, and aged proteins. The ability to degrade and recycle large damaged molecules via autophagy allows cells to survive under stress conditions, such as nutritional starvation, oxidative stress, hypoxia, endoplasmic reticulum (ER) stress, and metabolic stress [1]. Depending on the mechanism mediating the delivery of intracellular components to lysosomes, autophagy is classified as macroautophagy, microautophagy, and molecular chaperone-mediated autophagy. The delivery of autophagic cargo inside membrane vesicles to lysosomes for degradation is called macroautophagy [2]. The lysosomal membrane can internalize cargo into vesicles that are formed from the invagination of membranes. This process, known as microautophagy, contributes to the degradation of proteins and organelles, either in bulk or selectively [3]. Cytosolic proteins for degradation can also enter lysosomes via a protein translocation system at the lysosomal membrane, a process called molecular chaperone-mediated autophagy [4]. In addition, the maintenance of organellar quality is achieved by the selective elimination of organelles via autophagy, a process termed organelle phagy that includes pexophagy, mitophagy, and reticulophagy [5]. The most well-characterized form of autophagy is macroautophagy, which is also commonly referred to simply as autophagy. In this paper, we use the term autophagy to refer to macroautophagy, unless otherwise specified [6].

### 1.1. Molecular Mechanisms of Autophagy

Autophagy refers to the formation of autophagosomes by the cell membrane system to encapsulate part of the cytoplasm, organelles, or proteins that need to be degraded. After fusing with lysosomes to form autolysosomes, the contents of autophagosomes are degraded, so cellular homeostasis is maintained and organelles can be updated. Autophagy is regulated by the autophagy-related gene (ATG), which influences molecular signaling pathways [7]. Autophagic flux is the term used to describe the complete dynamic autophagy process. Evidence suggests that autophagy targets specific cargo, such as aggregated proteins, damaged mitochondria, excess peroxisomes, and invading pathogens, through a family of proteins known as autophagy receptors, which include p62/SQSTM1 (p62), optineurin (OPTN), neighbor of BRCA1 (NBR1), and nuclear protein 52 kDa (NDP52), which harbor both a Ub binding domain and an LC3 interaction region [8].

The autophagy pathway can be divided into four different stages: initiation, nucleation, maturation, and fusion and degradation (Figure 1).

#### 1.1.1. Initiation

The mammalian target of rapamycin (mTOR) is an evolutionarily conserved serine/threonine kinase that acts as a central negative regulator of autophagy. The protein acts as a sensor receptor for various intracellular signals, such as starvation, oxidative stress, energy stress, and pathogen infection, which directly or indirectly induce the activation of autophagy through the inhibition of mTOR. The protein kinase B/(PKB/AKT) pathway is activated by stress or pathogen infection and then phosphorylates tuberous sclerosis complex 2 (TSC2) and activates mTOR complex 1 (mTORC1), subsequently inhibiting autophagy [9,10]. By contrast, autophagy is activated by 5′-AMP-activated kinase (AMPK), which plays a key role in sensing cellular energy and ATP levels [11,12]. In response to stimuli or stress conditions, mTORC1 activates the unc-51-like autophagy-activating kinase (ULK)1/2 complex, which consists of ULK1 or ULK2 kinase, ATG13, FIP200, and ATG101, and suppresses the autophagy pathway [13,14]. In mammalian systems, autophagosome generation is initiated at multiple sites throughout the cytoplasm. Other signaling pathways are independent of mTOR, including the ER stress pathway, the inositol phospholipid signaling pathway, and so on [15].

#### 1.1.2. Nucleation

Following initiation, the membrane begins to expand and nucleate. At this stage, the vesicle is called a phagophore and is the primary double-membrane sequestering compartment. A class III phosphatidylinositol 3-kinase (PI3K) complex mediates phagophore formation [16]. The class III PI3K complex consists of three main components: VPS34, Vps15, and Beclin-1. The activity of this lipid kinase complex depends on various positive and negative regulators [17,18]. Beclin-1 is one of the key proteins in membrane nucleation, and its interaction with Bcl-2 can inhibit autophagy. Conversely, disruption of this interaction allows Beclin-1 to bind to lipid kinase VPS34 to promote membrane nucleation. Alternatively, Beclin-1 differentially modulates membrane formation by interacting with different components such as UV radiation resistance associated (UVRAG) [19], RUN domain Beclin-1-interacting and cysteine-rich domain-containing protein (RUBICON), ATG14L [18], Autophagy/beclin 1 regulator 1 (AMBRA1) [20], and Vacuole membrane protein (VMP1) [21]. VPS34-mediated enzymatic production of phosphatidylinositol 3-phosphate (PtdIns3P) provides a platform for phosphatidylinositol 3-phosphate (PI3P) recruitment of autophagy-related proteins, including WIPI1-4 and Double FYVE-containing protein 1(DFCP1), to promote the formation of autophagosomes [22,23].

#### 1.1.3. Maturation

The accumulation of PI3P-recruiting autophagy proteins at the membrane nucleation site results in the binding of additional ATGs, a step that is necessary for extension and closure of the autophagosome membrane. In the extension and autophagosome formation phase, there are two sets of ubiquitin-like conjugation systems: the ATG12-ATG5-ATG16L1 system and the LC3 system. The E1- and E2-like actions of ATG7 and ATG10 catalyze the covalent conjugation of ATG12 to ATG5. After this conjugation, ATG16L1 is recruited to form the ATG12-ATG5-ATG16L1 complex, which has an E3-like function as a second ubiquitin-like conjugation system [24,25,26]. By contrast, the LC3 system involves the combination of the LC3 protein and the lipid molecule phosphatidylethanolamine (PE) [27]. The LC3 precursor protein is cleaved by ATG4, which exposes the glycine residue from its carboxy terminus, resulting in PE binding. The activities of E1-like ATG7 and E2-like ATG3 proteins also lead to the conjugation of LC3-PE, also known as LC3-II, a hallmark of autophagy [7].

#### 1.1.4. Fusion and Degradation

After the autophagosome membrane has formed, autophagic vesicles are transported to the lysosome for degradation and autophagy-associated LC3 is degraded and recycled [28,29]. Several soluble N-ethylmaleimide-sensitive fusion protein attachment protein receptors (SNARE) proteins (including syntaxin 17 and vesicle associated membrane protein 8), lysosomal integrins, lysosomal-associated membrane protein 2 (LAMP2), and Ras-like GTPase (RAB) proteins play key roles in autophagosome-lysosome fusion [30]. As autophagosomes form autolysosomes via autophagosome fusion, their cargo is degraded by lysosomal proteases. Degradation products such as amino acids and fatty acids are then redirected to the cytoplasm for reuse in various metabolic processes.

### 1.2. Multiple Roles of Autophagy in Immune Responses

#### 1.2.1. Autophagy Plays a Role in Innate Immunity

The innate immune system promotes inflammatory responses to combat microbial infections by secreting inflammatory mediators. Autophagy is involved in the control of inflammation and is essential for proper functioning of the innate immune system [31]. RUBICON is a major regulator of autophagy and can terminate pathogen recognition receptor-induced cytokine production to prevent unchecked pro-inflammatory responses [32]. Autophagy directly regulates the secretion of inflammatory cytokines by inhibiting the inflammatory pathway [33], and knockout of LC3 and Beclin-1 promotes domain-like receptor protein 3 (NLRP3)-dependent inflammation through reactive oxygen species (ROS) accumulation and mitochondrial dysfunction [34]. Autophagy mediators also recognize ubiquitinated adipose-derived stromal cells (ASCs) and induce the selective degradation of inflammasomes, thereby inhibiting IL-1β and IL-18 production [31]. In addition to regulating the activation of NLRP3 inflammasomes, autophagy controls IL-1β production by targeting pro-IL-1β for lysosomal degradation [35]. Autophagy also regulates other pro-inflammatory signaling factors, in addition to the inflammasome [36], for example, by degrading Bcl-10 to reduce nuclear factor-κB (NF-κB) activation in antigen-activated T cells [37]. Some autophagy proteins participate in innate immunity, such as Beclin-1, ATG5, ATG7, ATG9, and ATG16L1 [38]. In some cell types, ATGs are able to inhibit the production of type I interferons (IFNs) and cytokines. For example, ATG5-ATG12 conjugation inhibits type I IFN via the recruitment of and interaction with domains of retinoic acid-inducible gene I (RIG-I), melanoma differentiation-associated protein 5 (MDA5), and interferon-beta promoter stimulator 1 (IPS-1) [39]. Similarly, in ATG5-deficient mouse embryonic fibroblasts, mitochondrial dysfunction elevates levels of reactive oxygen species (ROS) to enhance RIG-I-like receptor (RLR) signaling, leads to the production of large amounts of IFN-α and IFN-β, and inhibits vesicular stomatitis virus (VSV) replication. Neutralization of IFN-α and IFN-β reverses the VSV replication efficiency, eliminating differences in viral replication between wildtype and ATG5 knockout cells and demonstrating a role for autophagy in maintaining cellular homeostasis by clearing dysfunctional mitochondria, a mechanism that is required to prevent the accumulation of ROS and dysregulation of RLR pathways [40,41]. It has been reported that after binding to cyclic GMP-AMP (cGAMP), STING translocates to the ER-Golgi intermediate compartment (ERGIC), which acts as a membrane source for WIPI2 recruitment and LC3 lipidation, resulting in cytoplasmic DNA or viral DNA being targeted for degradation by the lysosome. These results have suggested that the induction of autophagy is a primordial function of the cGAMP synthase (cGAS)-STING pathway [42]. They have revealed a new antiviral pathway that functions via autophagy-regulated innate immunity.

#### 1.2.2. Autophagy Plays a Role in Adaptive Immunity

Autophagy plays a crucial role in the functions of major histocompatibility complex (MHC) class I and class II molecules, which recognize CD8^+^ T cells and CD4^+^ T cells, respectively [43,44]. Dendritic cells (DCs) from Crohn’s disease patients with NOD2 and ATG16L1 risk variants exhibit defective autophagy induction and MHC class II antigen presentation. Similarly, rapamycin-induced autophagy enhances the presentation of mycobacterial antigens and CD4^+^ T-cell responses in macrophages. Intracellular antigens captured by autophagosomes can be degraded by amphisomes (produced by autophagosomes and endosomes) and loaded onto MHC class I molecules for presentation to CD8^+^ T cells. Pathogens such as herpes simplex virus type 1 (HSV-1) can initiate this process, triggering the processing and presentation of endogenous viral antigens on MHC class I molecules. In addition, antigen-presenting cells such as DCs can process MHC class I-presented extracellular antigens via cross-presentation, a process that also relies on autophagy. Studies support the view that antigen-presenting cells can switch the classical type I and type II MHC presentation pathways, allowing information exchange between intracellular organs, to mount an effective immune response to endogenous and exogenous antigens [45,46]. MHC class II antigens were originally thought to be microbial materials from outside the cell derived from endocytosis or phagocytosis and lysosome degradation. However, it has been shown that more than 2% of the natural ligands eluted by affinity purification of MHC class II molecules are derived from endogenous proteins under inflammatory conditions, and autophagy plays a role in their production [47]. Therefore, autophagy may subject cells under potentially dangerous stress conditions to enhanced immune surveillance by CD4^+^ T cells.

## 2. Modulation of Autophagy by DNA Viruses

### 2.1. A Brief Introduction to DNA Viruses

DNA viruses include double-stranded (ds) DNA viruses, single-stranded (ss) DNA viruses, and dsDNA/ssDNA viruses. Of these, dsDNA viruses are the most diverse, including 1737 different types that account for 37.12% of all known viruses. ssDNA viruses include 856 types that account for 18.29% of all known viruses. However, there are only two types of dsDNA/ssDNA chimeric viruses [48]. In total, 25 families are represented by dsDNA viruses, including many that infect humans or animals, such as *Adenoviridae*, *Ascoviridae*, *Asfarviridae*, *Baculoviridae*, *Poxviridae*, *Polyomaviridae*, *Papillomaviridae*, *Nimaviridae*, *Iridoviridae*, and *Herpesviridae*. The range of hosts infected is wide and includes both vertebrates and invertebrates, such as insects and shrimp. Seven virus families are represented by ssDNA viruses, including some that infect humans or animals, such as *Anelloviridae*, *Circoviridae*, and *Parvoviridae* [49]. Autophagy is a process by which cells maintain homeostasis in the intracellular environment, where viral infections can cause disorders. When cells are stimulated by a virus, interaction between the virus and autophagy may be triggered. Autophagy is a component of innate immunity and has an antiviral effect; however, viruses often encode proteins to evade and resist this process. Alternatively, viruses can use autophagy to enhance their own replication or increase the persistence of latent infections. The known interactions between animal or human DNA viruses and the cellular autophagy machinery are shown in Table 1. Given these known interactions, how does autophagy affect DNA viral replication? The pro- and antiviral effects of autophagy during DNA viral infection are of great interest, particularly for significant human and animal pathogens such as Epstein–Barr virus (EBV) and porcine circovirus type 2 (PCV2). In the following sections, we discuss recent progress in our understanding of the role of autophagy in DNA viral replication (Figure 2) and the mechanisms by which autophagy regulates viral replication (Figure 3).

### 2.2. Herpesviridae

The earliest and most comprehensive research has been on the relationship between Herpesviridae family viruses and autophagy. The first anti-autophagic protein identified was ICP34.5, a neurovirulence protein found in HSV-1, a member of the Alphaherpesvirinae subfamily of the Herpesviridae family [89]. ICP34.5 binds to the key autophagy regulator Beclin-1 to inhibit autophagy, and viruses lacking the ICP34.5 gene trigger autophagy by activating the eukaryotic translation initiation factor 2-kinase 2 (eIF2AK2)/protein kinase RNA-activated (PKR) pathway [90]. It is thought that PKR contributes to the regulation of starvation-induced autophagy. In addition, during the HSV-1 infection process, the US11 protein, which is expressed later than ICP34.5, can directly bind to PKR to inhibit autophagy, but is unable to inhibit autophagy in PKR-deficient cells [91]. A further study showed that disruption of the tripartite motif protein 23 -TANK-binding kinase 1 complex by the US11 protein inhibits autophagy-mediated restriction of HSV-1 infection [92]. Autophagy activation protects adult mice from encephalitis during HSV-1 infection, but is associated with increased apoptosis in the brains of newborn mice infected with HSV-1. Therefore, the level of protection against HSV-1 infection provided by autophagy differs, depending on the age of the animal [53]. It is worth noting that autophagy does not have the same effect on HSV-1 infection in all cell types. Autophagy is critical for the control of HSV-1 in primary neurons, but is not necessary in fibroblasts [54]. Similarly, autophagy has antiviral effects in nerve cells, but not epithelial cells [90]. In fibroblasts, nerve cells, and epithelial cells, ICP34.5 inhibits autophagosome initiation. However, the effect of ICP34.5 on autophagy differs in different antigen-presenting cells. This mutant virus is highly neuroattenuated in vivo, suggesting that the ICP34.5-mediated blockade of Beclin-1-dependent autophagy is required for neurovirulence [90]. In DCs, ICP34.5 blocks the maturation of autophagosomes, which contributes to immune evasion by reducing DC antigen presentation [55,93]. Although the regulation of autophagy by HSV-1 depends on cell type, most researchers have found that autophagy adversely affects HSV-1 infection by increasing antigen presentation or reducing viral replication. However, some studies have found that HSV-1 infection transiently induces autophagy in human mononuclear THP-1 cells and this autophagy plays a role in promoting viral infection [94]. Indeed, pretreatment of cells with autophagy inhibitors before infection with a high multiplicity of infection (MOI) of HSV-1 results in a reduced viral titer [45]. Similarly, Beclin-1 knockdown in THP-1 cells reduces HSV-1 replication. Autophagy may also facilitate viral entry into cells, but this mechanism remains to be studied [94]. In summary, HSV-1 ICP34.5 and US11 proteins mediate the blockade of autophagy to evade and resist antiviral mechanisms and increase virus survival. In addition, a novel strategy employed by HSV-1 to evade the host during the early stages of infection is the down-regulation of p62 and OPTN mediated by viral infected cell protein 0 (ICP0). Cells lacking p62 or OPTN produce greater antiviral responses, while cells expressing exogenous p62 show reduced viral yields [95].

HSV-2 also has genes encoding ICP34.5 and US11 proteins. However, the relationship between these two proteins and autophagy during HSV-2 infection has not been studied. Similar to HSV-1, autophagy appears to be inhibited in HSV-2-infected fibroblasts, but in contrast to the HSV-1 homolog, the HSV-2 major neurovirulence factor ICP34.5 is encoded by a spliced gene that contains an intron. In addition, the N-terminal domain of HSV-1 ICP34.5, which binds Beclin-1 and TBK1, has only some sequence homology to HSV-2 ICP34.5, and an insertion appears to disrupt the corresponding Beclin-1 and TBK1 structures in the HSV-2 ICP34.5 domain. Therefore, the autophagy-inhibiting viral protein has yet to be identified [56]. Treatment of cells with the autophagic flux inhibitor bafilomycin reduces HSV-2 replication, indicating that autophagy is beneficial to viral replication in fibroblasts. Reductions in the viral replication efficiency observed in ATG5-deficient cell lines also confirm this view. However, the mechanism by which HSV-2 uses autophagy to enhance its replication requires further study [56].

Varicella zoster virus (VZV) is a member of the *Alphaherpesvirinae* subfamily and is the etiologic agent responsible for varicella and herpes zoster. Several studies have shown that VZV infection triggers autophagy and does not interfere with this cellular response in the same way as HSV-1 does, presumably because the virus does not encode viral gene products capable of preventing autophagosome formation [57,96,97]. Wild and attenuated strains of VZV induce autophagy in different cell types during the late stages of infection. High numbers of autophagosomes are observed in live tissue sections from patients infected with VZV. Inoculation of immunodeficient mice with xenografted human skin with VZV results in the accumulation of LC3-positive autophagosomes [59]. Furthermore, pulse-tracking experiments using a dual fluorescent LC3 reporter plasmid confirmed that the virus induces autophagy in infected fibroblasts. The mechanism by which autophagy is induced during VZV infection is not well-understood. One possibility is that VZV may trigger ER stress to maintain cellular homeostasis. Evidence of ER stress during VZV infection includes ER swelling and initiation of the unfolded protein response (UPR), which is induced to relieve ER stress [97]. By contrast, a recent study demonstrated that both rOka and vOka strains of VZV inhibit the late stages of mTOR-mediated autophagic flux (either autophagosome-lysosome fusion or subsequent degradation). Cell-associated rOka shows more significant inhibition than vOka, especially when cells are under starvation conditions [58]. To demonstrate that autophagy promotes VZV replication, Buckingham et al. [98] used different strategies to modulate autophagy and analyzed viral parameters such as infectivity and viral protein expression. First, they treated cells with 3-methyladenine (3-MA) to inhibit autophagy. The role of 3-MA is controversial as it has also been shown to promote autophagy in cells after prolonged treatment [99]. However, treatment of VZV-infected human melanoma cells with 3-MA and ATG5-targeting small interfering RNAs (siRNAs) reduced virus proliferation and decreased viral infectivity. In addition, 3-MA treatment decreased the expression of the VZV glycoprotein gE, whereas the autophagy activator trehalose increased its expression, and the molecular weights of gE and gI were decreased in cells lacking ATG5 compared to normal cells. This shows that viral glycoprotein synthesis has a minor effect on autophagy-deficient cells, leading to the accumulation of gE dimers. Because viral glycoprotein synthesis induces ER stress, one hypothesis is that autophagy alleviates ER stress and allows glycoproteins to be correctly synthesized. Another study revealed the co-localization of gE, LC3, and RAB11, a marker of recycled endosomes, and researchers using immunoelectron microscopy found that vesicles containing enveloped virions were only monolayers and therefore did not exhibit the characteristics of autophagosomes. This suggests that some viral particles accumulate in a heterogeneous population of single-membraned vesicular compartments after secondary envelopment and become labeled with components from both the endocytic pathway (Rab11) and the autophagy pathway (LC3). The authors of this study proposed that these vesicles may be autophagic endosomes containing only one or several virions and may be used to release vesicles from the cell [100]. Bafilomycin A1, an anti-autophagy antibiotic, disrupts the secondary envelopment site of the VZV capsid by altering the pH of the trans-Golgi network, thereby preventing proper formation of the viral assembly compartment [101].

Duck enteritis virus (DEV) belongs to the *Alphaherpesvirinae* subfamily in the *Herpesviridae* family and causes an acute, septic, sexually transmitted disease in ducks, geese, and other poultry. We found that DEV infection triggers autophagy in duck embryo fibroblast (DEF) cells, as demonstrated by the appearance of autophagosome-like double- or single-membrane vesicles in the cytoplasm of host cells and a number of green fluorescent protein (GFP)-tagged LC3 dots. In addition, increased conversion of LC3I and LC3II and decreased p62/SQSTM1 indicated complete autophagic flux [60]. DEV infection for 48 h induces a decrease in cellular ATP levels. We found that DEV infection activated the metabolic regulator AMPK and inhibited mTOR activity. In cases in which AMPK inhibited mTOR and downregulated autophagy and DEV replication, AMPK expression was not changed. However, siRNA targeting of AMPK inhibited the activation of tuberous sclerosis 2 (TSC2). Therefore, our findings indicate that energy metabolism in cells damaged by DEV contributes to autophagy via the AMPK-TSC2-mTOR signaling pathway [102]. In addition, ER stress is triggered by DEV infection, as demonstrated by the increased expression of the ER stress marker glucose-regulated protein 78 (*GRP78*) gene and dilated morphology of the ER. Pathways associated with the UPR, including the PKR-like ER protein kinase (PERK) and inositol-requiring enzyme 1 (IRE1) pathways, but not the transcription factor 6 (ATF6) pathway, are activated in DEV-infected DEF cells. In addition, the knockdown of both PERK and IRE1 by siRNA reduced levels of LC3II and viral yields, suggesting that the PERK-eukaryotic initiation factor 2α (eIF2α) and IRE1-x-box protein 1 (XBP1) pathways may contribute to DEV-induced autophagy [103]. Wortmannin and LY294002 inhibit the PI3K pathway and the early stages of autophagy. In DEF cells treated with wortmannin or LY294002 for 4–6 h and then infected with DEV at different time points, the conversion of LC3I to LC3II was decreased, indicating that autophagy was inhibited. Meanwhile, the number of progeny viruses recovered was lower in wortmannin- or LY294002-treated infected cells than in mock cells at various time points. To exclude any nonspecific effects of pharmacological components, siRNA was used to interfere with the expression of Beclin-1, which is a very important regulator of autophagosome formation and maturation, and ATG5 in DEV-infected cells. In these experiments, no endogenous Beclin-1 or ATG5 protein was detected via Western blotting, and the conversion of LC3I to LC3II was reduced. The number of progeny viruses collected from siRNA-transfected cells was also higher than from mock cells at 48 h. To further explore the relationship between autophagy and DEV replication, DEF cells were treated with rapamycin, which inhibits the activity of mTOR and promotes cell autophagy. The results showed a higher LC3II/LC3I ratio in rapamycin-treated DEV-infected cells than in untreated DEV-infected cells, indicating that rapamycin promotes autophagy in infected cells. The number of progeny viruses collected from rapamycin-treated cells was also higher than from mock cells at various time points [60].

Pseudorabies virus (PRV) is a swine herpesvirus in the *Alphaherpesvirinae* subfamily with a broad host range that causes devastating disease in infected pigs. During the early infection period, PRV virions induce autophagy without viral replication, whereas PRV reduces the basal level of autophagy in several permissive cell types when viral proteins are expressed. Furthermore, Sun et al. [61] found that the PRV tegument protein US3 inhibits the autophagy response by activating PI3K/AKT pathways. It has been reported that rapamycin induces autophagy, which reduces PRV replication, whereas the inhibition of autophagy by 3-methyladenine (3-MA) and knockdown of endogenous *ATG5* and *LC3B* promotes PRV replication [61]. However, Xu et al. [62] found that PRV induced autophagy via the classical Beclin-1-ATG7-ATG5 pathway to enhance viral replication in N2a cells in vitro. Because the infection doses used in the two studies were different, the high MOI may be an important reason why Sun et al. did not observe the induction of autophagy after PRV infection.

Human cytomegalovirus (HCMV) is a beta-herpesvirus and a ubiquitous opportunistic pathogen. Like other viruses in the *Herpesviridae* family, HCMV can persist in the host in an inactive state known as latency after primary infection. Increases in autophagosomes and autophagic flux are induced during the early stages of HCMV infection, independent of viral protein synthesis, as ultraviolet-inactivated HCMV still promotes autophagy. At later infection time points, HCMV inhibits autophagy in cells through the expression of viral proteins. In addition, both the internal repeat sequence 1 (IRS1) and terminal repeat sequence 1 (TRS1) of HCMV are able to block autophagy in different cell lines, independent of the eukaryotic translation initiation factor 2 subunit alpha (eIF2S1) kinase eIF2AK2/PKR (67). Instead, TRS1 and IRS1 interact with the autophagy protein Beclin-1. Similarly, it was found that in HCMV-infected cells transfected with GFP-LC3, the number of punctate structures increased from 24 to 48 h, but decreased at 60 h after infection. Furthermore, monodansylcadaverine and acridine orange staining suggested that autophagy was induced during early stages of infection, but was inhibited at later stages, in response to HCMV infection [63]. Similarly, mouse cytomegalovirus (MCMV), a common model of HCMV, induces autophagy during the early stages of infection and then blocks it. Studies on retinal pigment epithelial (RPE) cells revealed the presence of autophagic vacuoles during the early stages of MCMV infection. Conversely, autophagic flux was increased at 24 h after infection. These results indicate that MCMV may employ strategies to inhibit or block the later stages of autophagy, such as the formation of autophagosomes or the degradation of their contents [65]. Studies have revealed that the autophagy inducer rapamycin is beneficial to HCMV replication, whereas the treatment of cells with the autophagy inhibitor Spautin-1 reduces viral titers. These results indicate that HCMV may use the autophagy process to promote its own replication. However, they contradict a study indicating that trehalose can inhibit HCMV replication by triggering autophagy [64]. Therefore, the specific relationship between HCMV and autophagy requires further investigation [63].

Kaposi’s sarcoma-associated herpes virus (KSHV) belongs to the *Rhadinovirus* genus in the human *Gammaherpesvirinae* subfamily and is also called HHV8. KSHV encodes several proteins that mimic cellular homologs. Among these proteins, viral Bcl-2 and the viral Fas-associated death domain-like IL-1β-converting enzyme-like inhibitory protein (FLIP) are homologs of cellular Bcl-2 and cellular FLIP, respectively. In addition to preventing cell death, these proteins have a strong influence on autophagy. For example, viral FLIP inhibits autophagy by preventing ATG3 from binding to and processing LC3 [66]. Researchers have found that the KSHV replication and transcriptional activator (RTA) induces autophagy and promotes the KSHV lytic cycle [67], but blocks the final stage of autophagy. Experiments in which this molecule was silenced demonstrated that the downregulation of Ras-related protein Rab-7a (RAB7) is one of the mechanisms leading to autophagy blockade [68]. It has also been reported that the KSHV cleavage protein K7 blocks autophagosome maturation through interactions with Rubicon during the KSHV lytic cycle [104]. Other KSHV proteins expressed during the lytic cycle, such as the viral G-protein coupled receptor (GPCR), may also negatively regulate autophagy through activation of the PI3K/AKT/mTOR pathway [105]. 3-MA has also been used to suppress autophagy pathways in Vero cells latently infected with KSHV. It was demonstrated that 3-MA reduces KSHV lytic reactivation, especially during the early phase, and that this reduction is likely due to the effect of 3-MA on autophagy (73). In addition, the inhibition of autophagy was affected by RTA-mediated lytic replication [67].

EBV is a gammaherpesvirus capable of establishing latent infections and causing many epithelial and lymphoid malignancies. One study showed that the lytic protein RTA induces autophagy by activating extracellular signal-related kinase 1/2 (ERK1/2) [106]. It has also been reported that the latent membrane protein (LMP)1, a viral oncogene, and LMP2A, which induces cancer transformation, migration, invasion, and differentiation, can both activate autophagy during latency [107,108]. Furthermore, it has been shown that the final stage of autophagy is inhibited by EBV, which protects the virus from elimination by lysosomal proteases [69]. In investigations of the influence of autophagy on EBV replication, ATG12 and ATG16 silencing reduced the production of EBV particles in 293/EBV wildtype cells by 60%–80% during lytic EBV infection. Conversely, infectious particle production was increased upon the pharmacological stimulation of autophagy [69]. These results suggest that autophagic membranes might contribute to the cytosolic maturation of EBV. In addition, LC3II is present in purified EBV viral particles, indicating that LC3II is incorporated into the EBV virus during its maturation in the cytosol and strongly suggesting that autophagic membranes contribute to the final envelope of this human tumor virus [69]. Because inhibition of the early stages of autophagy impairs EBV replication and viral particles have been observed in autophagic vesicles in the cytoplasm of producing cells, EBV appears to be transported using autophagy mechanisms, resulting in enhanced viral replication [69]. In summary, inhibition of the early stages of autophagy or the use of strategies to overcome autophagy blockade allows for the degradation of viruses and membranes from autophagosomes in lysosomes as part of the EBV replication cycle [70]. One novel study showed that p62-mediated selective autophagy is constitutively induced during EBV latency and correlates with ROS-Keap1-NRF2 pathway activity, demonstrating its crucial role in the regulation of the response to DNA damage during viral latency [109].

Murine gammaherpesvirus 68 (MHV68) causes lymphocytic disease in mice. After an initial acute productive infection, MHV68 persists in mice by establishing latency in peritoneal macrophages and splenic B cells, regardless of the route of infection [110]. During latency, MHV68 M11 blocks autophagy by interacting with Beclin-1. Moreover, inhibition of autophagy by M11 plays a role in the maintenance of latency, although it has no appreciable impact on the establishment of latency [72].

### 2.3. Adenoviridae

Oncolytic adenovirus is a type of genetically engineered adenovirus that selectively replicates and lyses tumor cells. Oncolytic adenoviruses induce autophagy in tumor cells, and the proteins adenovirus early 1A (E1A) and E1B play an important role in the signaling pathways that induce autophagy [51]. Adenovirus E1A binds to the retinoblastoma (RB) tumor suppressor, which results in the release of the E2F transcription factor 1 (E2F1) from the RB-E2F1 complex. E2F1 activation induces autophagy by upregulating the autophagy-related proteins ATG5 and LC3. By contrast, E1B competes with Bcl-2 to form part of the Beclin-1 interactome, leading to separation of the Beclin-1-Bcl-2 complex and the induction of autophagy [111]. In contrast to the inhibition of autophagy that occurs when Beclin binds to HSV-1 ICP34.5, the removal of Bcl-2, a negative regulator of autophagy, and integration of E1B into the Beclin complex, favor the interaction between Beclin-1 and PI3KCIII, which in turn results in the formation of autophagosomes. When autophagy is induced after viral infection, oncolytic adenoviruses also induce the lysis of tumor cells, leading to viral expansion. Since high levels of autophagy induce cell lysis, autophagy inhibition strategies can be exploited to extend the lifespan of tumor cells and inhibit viral spread [50].

### 2.4. Papillomaviridae

Human papilloma virus (HPV) is a dsDNA virus with a circular genome that belongs to the *Papillomaviridae* family and can cause cervical cancer. According to one study, HPV virions are coated with heparan sulfate proteoglycans that interact with epidermal growth factor receptors (EGFRs) present on the plasma membrane of target cells and PKB/AKT and the phosphatase and tensin homolog (PTEN), resulting in the phosphorylation and activation of mTOR. Activated mTOR can then inactivate ULK1, thereby inhibiting autophagosome formation [78,112,113]. Like many other viruses, HPV manipulates autophagy to promote the life cycle of virus-infected host cells. By inhibiting autophagy, the oncogenic HPV virus promotes binding and internalization to support the proliferation of infected epithelial cells, significantly promoting cancer progression [78]. HPV16, a high-risk HPV, is a carcinogenic virus with the ability to control autophagy throughout its life cycle. There is significant evidence that HPV16 viral proteins inhibit autophagy through various mechanisms at every step of the infection and during the different tumorigenesis processes by targeting different stages in the autophagy pathway. However, there has been little research on the similarities and differences between low- and high-risk serotype infections in terms of their role in autophagy.

### 2.5. Circoviridae

PCV2 belongs to the genus *Circovirus* in the *Circoviridae* family and is the main cause of porcine circovirus-associated disease. Its genome is a single strand of covalently closed circular DNA of approximately 1.7 kb in size. A study showed that in PCV2-infected PK-15 cells, AMPK and extracellular signal-related kinase (ERK)1/2 negatively regulate the mTOR pathway by phosphorylating TSC2; therefore, PCV2 may induce autophagy via the AMPK/ERK/TSC2/mTOR signaling pathway [76]. Ochratoxin A (OTA) induces autophagy in PK-15 cells and promotes PCV2 replication, both in vivo and in vitro. Inhibition of autophagy by 3-MA and chloroquine, which inhibits autophagy by inhibiting lysosomal activity during the late stages, significantly attenuates OTA-induced PCV2 replication [114]. These results were confirmed by the siRNA knockdown of ATG5 and Beclin-1. Further experiments showed that *N*-acetyl-l-cysteine, an ROS scavenger, can block autophagy induced by OTA, indicating that ROS may be involved in the regulation of OTA-induced autophagy [114]. In addition, the administration of 75 μg/kg OTA also significantly increased PCV2 replication and autophagy in the lungs, spleen, kidney, and bronchial lymph nodes of pigs [76]. Taken together, these results indicate that OTA-induced autophagy promotes PCV2 replication in vitro and in vivo. In addition, autophagy protects host cells from potential apoptosis and promotes PCV2 replication via oxidative stress, which may partially explain the pathogenesis of PCV2 associated with oxidative stress-induced autophagy [75].

### 2.6. Parvoviridae

Human parvovirus B19 (B19) belongs to the genus *Erythroparvovirus* in the *Parvoviridae* family. Its genome consists of 5.6 kb of ssDNA. Immunofluorescence confocal microscopy has been used to observe endogenous LC3 staining to determine whether autophagosome formation is induced by the virus. It was found that B19 infection induces cellular autophagy. The ratio of LC3I to LC3II is also significantly increased in infected cells, suggesting that autophagy may be involved in the B19 infection process [77].

### 2.7. Poxviridae

The vaccinia virus is a member of the genus *Orthopoxvirus* in the *Poxviridae* family. Evidence suggests that the vaccinia virus selectively inactivates cellular autophagy machinery through a novel molecular mechanism, the conjugation of ATG12 to ATG3, and consequent aberrant LC3 lipidation [74]. Infection with oncolytic vaccinia, in which the viral thymidine kinase gene is insertionally inactivated, increases autophagy in human hepatocellular carcinoma MHCC97-H cells by stimulating the ER stress-induced signaling pathway [73]. 

### 2.8. Polyomaviridae

The John Cunningham virus (JCV) is a human neurotropic polyomavirus that causes the fatal demyelinating disease progressive multifocal leukoencephalopathy. One study showed that overexpression of the Bcl-2-related athanogene (Bag) protein family member Bag3 significantly reduced expression levels of T antigen by inducing autophagy. Interestingly, this resulted in the inhibition of JCV infection by glial cells, indicating that the overexpression of Bag3 has an impact on the viral lytic cycle [82].

The human BK polyomavirus (BKPyV) is a small double-stranded DNA virus that can cause tumors. BKPyV infection is not only associated with autophagosome formation, but also with viral particle localization to autophagy-specific compartments during the early stages of infection. One study reported that excess amino acids reduced BKPyV infection by inhibiting autophagy. The inhibitors 3-MA, bafilomycin A1, and spautin-1 also disrupted autophagy and reduced viral infection, whereas treating cells with rapamycin increased infection. Furthermore, siRNA knockdown of the autophagy genes ATG7 and Beclin-1 led to a reduction in BKPyV infection. These data support a role for autophagy in promoting BKPyV infection [84].

### 2.9. Asfarviridae

African swine fever virus (ASFV) is the only member of the family *Asfarviridae*, genus *Asfivirus*. In August 2018, Liaoning Province experienced the first African swine fever epidemic in China. Treatment with starvation and rapamycin before viral infection reduces the number of cells that are subsequently infected, but how this process changes viral infection remains unclear. One study showed that ASFV does not induce autophagy in infected cells via LC3 activation and/or autophagosome formation. It was further confirmed that ASFV A179L, a vBcl2 homolog, interacts with Beclin-1 to control autophagy [79].

### 2.10. Hepadnaviridae

Hepatitis B virus (HBV) is a strictly hepatotropic DNA virus that belongs to the *Hepadnaviridae* family. Many published studies have reported that HBV can induce autophagy in vitro and in vivo [115,116,117]. The key roles of autophagy in the normal life cycle of HBV, such as formation of the HBV envelope, have gradually been revealed by researchers in different laboratories [80,118]. HBV triggers the autophagy pathway, which activates viral DNA replication, and HBV proliferation is suppressed upon the inhibition of autophagy [80]. In addition, in chronic HBV infections, early autophagy can be conducive to HBV replication, thereby worsening liver infection [81]. Furthermore, hepatocellular carcinoma (HCC) has been shown to be closely related to HBV-induced autophagy, as autophagy is lower in HBV-induced HCC than in simple chronic HBV infection, and increased autophagy may suppress HBV-induced HCC tumorigenesis. Therefore, autophagy activation can attenuate tumor progression through autophagic cell death and anti-tumor immune responses [81,119].

In summary, several DNA viruses have developed strategies to escape autophagy via the expression of specific anti-autophagy proteins, such as HSV-1, ASFV, and MHV68. However, there is increasing evidence to suggest that DNA viruses not only inhibit autophagy, but also hijack it. This benefit is obvious, because the inhibition of autophagy will result in reduced titers of viruses such as HCMV, EBV DEV, and PCV2. Conversely, the stimulation of autophagy will improve viral proliferation. Some viruses block the maturation of autophagosomes to avoid degradation, and autophagosome membranes are then used to promote the encapsulation and/or excretion of viral particles. During incubation, autophagy can also be activated by latent proteins encoded by different carcinogenic herpesviruses to promote cell survival and achieve long-term viral persistence in vivo. Therefore, the DNA virus role is more that of a “robber” than a “fugitive”. However, some DNA viruses and autophagy interactions and mechanisms need further research, such as those in the *Poxviridae* and *Papillomaviridae* families.

## 3. Role of Autophagy in Immune Responses to DNA Virus Infection

It has been shown that autophagy plays an important role in both innate and adaptive immunity. It contributes to innate immunity by participating in the capture, degradation, and elimination of intracellular viruses and to adaptive immunity by preparing endogenous and exogenous antigens for MHC class I and MHC class II presentation [120] (Figure 4).

### 3.1. Role of Autophagy in Innate Immune Responses to DNA Virus Infection

Viral contact with the cell surface may trigger the rapid activation of autophagy pathways through Toll-like receptor (TLR) recruitment of myeloid differentiation factor 88 (MyD88). A study showed that the ectopic expression of MyD88 significantly increased the number of GFP-LC3 puncta. By contrast, HSV-1 infection does not trigger autophagy in MyD88-deficient cells. Moreover, autophagic inhibitors have no effect in the absence of MyD88. These findings suggest that MyD88 is one of the key factors involved in the signaling pathway leading to autophagy activation by HSV-1 in monocytic cells [94]. In HSV-1 infected cells, the autophagy-related gene Beclin-1 inhibits DNA stimulation by interacting with cGAS, impeding cGAMP synthesis and halting IFN-I production. In addition, autophagy-mediated degradation can clear cytosolic pathogen DNA detected by DNA sensors, thereby indirectly attenuating the cGAS-STING signaling pathway [121].

MHV68 infection is characterized by latency in macrophages, and its reactivation is inhibited by IFN-γ. Park et al. [122] used a lysozyme-M-cre expression system to delete *ATG* genes, including *ATG3*, *ATG5*, *ATG7*, Beclin-1, and *ATG16*, in the myeloid compartment and showed that this inhibited the activation of MHV68 in marrow-derived macrophages. The effect of viral activation in ATG5-deficient macrophages cannot be explained by changes in viral replication or the establishment of latent infection. Instead, chronic infection with MHV68 increased systemic inflammation and IFN-γ production and resulted in an IFN-γ-induced transcriptional signature in macrophages of mice with several *ATG* deletions. However, the deletion of *ATG5* did not alter the activation of MHV68 in splenic B cells. Therefore, ATGs inhibit viral-induced systemic inflammation in macrophages, creating an environment that promotes the MHV68 transition from latency to activation [122].

The anti-KSHV viral GPCR-induced tumorigenesis mechanism in epithelial cells may be related to decreased viral GPCR protein levels, which subsequently inhibit cancer-promoting IL-6 signals. The autophagy protein Beclin-2 is essential for the endolysosomal degradation of certain cellular GPCRs, and Beclin-2 knockout mice exhibit deficiencies in autophagy and GPCR degradation. It has been demonstrated that lysosomal-dependent Beclin-2 reduces viral GPCR levels and viral GPCR-induced IL-6 signaling in vitro. At the same time, viral GPCR expression is enhanced and IL-6 production increased in mice lacking the Beclin-2 gene in vivo [123]. Furthermore, IL-6 elevation is negatively correlated with survival time in KSHV-infected *BECN2*^+/−^ mice. Several studies have suggested that IL-6 is a key virulence factor in KSHV-associated malignancies [124,125]. Although a causal relationship between increased IL-6 production and accelerated tumor formation and lethality in *BECN2*^+/−^ mice has not been established, the results are consistent with the conclusion that IL-6 is a key regulator of KSHV pathogenicity and provide clear evidence that Beclin-2 regulates KSHV GPCR-induced IL-6 levels in mice [125].

The MCMV protein M45 binds to the NF-κB essential modulator/inhibitor of NF-κB kinase subunit gamma (NEMO/IKKγ), a key regulator of NF-κB signaling, and is delivered to autophagosomes and transported to lysosomes for degradation. This results in attenuation of the host inflammatory response, allowing the virus to use autophagy to escape the immune system [65].

Oncolytic adenovirus-induced autophagy contributes to the induction of immunogenic cell death and subsequent release of damage-associated molecular pattern molecules such as adenosine triphosphate (ATP), high-mobility group box 1 protein (HMGB1), calreticulin (CRT), and uric acid [50].

### 3.2. Autophagy and Adaptive Immunity Responses to DNA Viruses

It is known that autophagy plays important roles in antiviral innate immunity and in adaptive immunity. A previous study showed that xenophagy can reduce the neurovirulence of HSV-1 and that *ICP34.5* knockout prevents HSV-1 from inhibiting autophagy, leading to reduced infection in the cornea of infected mice [126]. However, knockout of the Beclin-1-binding domain had no effect on infection in mice lacking B and T cells, indicating that protective effects are mediated by adaptive rather than innate immunity [123]. Consistent with these results, mice infected with mutant HSV-1 virus had higher virus-specific CD4^+^ T-cell responses compared to animals infected with wildtype virus. These data showed that autophagy can reduce HSV-1-mediated disease via adaptive immunity [126].

A study has shown that the potential immune evasion mechanism of HSV-1 may be related to its ability to inhibit autophagy in antigen-presenting cells via ICP34.5. Data also suggest that ICP34.5 does not inhibit the induction of autophagy in DCs [127], but does interfere with autophagy in nerve and fibroblast cells [93,96,128]. Another study demonstrated a strong interaction between ICP34.5 and its Beclin-1-binding domain that led to the suppression of autophagy in DCs and increased MHC class II presentation [93]. English et al. [45] used a model system with a paraformaldehyde-fixed macrophage cell line and β-galactosidase-inducible HSV-1 gB-specific CD8^+^ T-cell hybridomas to demonstrate that autophagy-mediated viral antigen processing may be involved in HSV-1 MHC class I presentation. Furthermore, they also found that the inhibition of HSV-1 ICP34.5-mediated autophagy abrogated endogenous viral antigen presentation to MHC class I molecules in primary DCs [46]. One study examined hematopoietic cells from transgenic mice lacking the *ATG5* gene and mice with reconstituted hematopoietic compartments, including cDCs and T cells in the spleen, lymph nodes, and bone marrow, that were infected with HSV-1. After 7 days, IFN-γ secretion from CD4^+^ T cells and the number of cells secreting IFN-γ in *ATG*^-−^ hematopoietic cells were significantly reduced compared to the wildtype control. These findings provide in vivo evidence that autophagy mechanisms are important for the presentation of multiple antigenic forms by MHC II molecules in DCs and are of great importance in vaccine design [129].

It was found that short infectious cell culture protein 10 (ICP10) peptides derived from HSV-2 are a viral inducer of protein aggregation (VIPA) that can convert a soluble green fluorescent protein into an aggregation-prone protein. VIPA can efficiently guide surrogate tumor antigens to the autophagosome/lysosomal degradation pathway, thereby significantly increasing MHC class I and class II antigen presentation. Simultaneously, MHC class I and class II molecules of the immune response of T cells to tumor antigens were induced to effectively protect immune animals from tumor challenges [130].

To conclude, given that autophagy plays a crucial role in the immune response, it has attracted great interest as a therapeutic target. However, the conflicting roles of autophagy in immunity and in a variety of biological processes complicate its therapeutic applications.

## 4. Conclusions

The interaction between DNA viruses and autophagy is complex. There are viruses that induce autophagy, viruses that inhibit autophagy, viruses that induce and then inhibit autophagy, and viruses that induce incomplete autophagy. The relationship between viruses and autophagy depends on the strain, cell type, and infection dose. Although the relationships between most DNA viruses and autophagy have been explored, some studies still need to be performed. For example, contradictory results have been found regarding the relationships between PRV and autophagy and between VZV and autophagy. In addition, most studies have been conducted in vitro, and the results from in vivo experiments need to be further explored. Furthermore, the mechanisms underlying viral suppression or induction of autophagy require further study.

The mechanisms of autophagy are used by some viruses to accelerate infection. However, autophagy can also have an antiviral effect through its role in innate immunity. Because autophagy has these two functions, specific methods are required for different viruses to either activate or inhibit autophagy and/or regulate the degree of autophagy in cells. This may aid in the development of novel strategies to control viral infections, especially for viruses that do not currently have suitable drug targets, and provide an avenue for the development of antiviral drugs in the future. However, how autophagy specifically inhibits or promotes viral replication and at which stages requires further study.

The interactions between viruses and cells are extremely complicated. Autophagy is not an isolated event and is always accompanied by various cellular changes. Autophagy can strongly protect certain viruses by regulating host immune pathways. In some cases, autophagy modulators not only directly enhance antiviral activity against DNA viral infections, but also enhance immune responses and vaccinations. Therefore, targeted autophagy may lead to a new class of specific antiviral therapies for the treatment of DNA virus-associated diseases. It may be worthwhile to explore whether pharmacological modulators and other molecules affect autophagy-related genes or pathways and can be converted into useful therapeutic agents for viral infections in humans or animals.

## Figures and Tables

**Figure 1 viruses-11-00776-f001:**
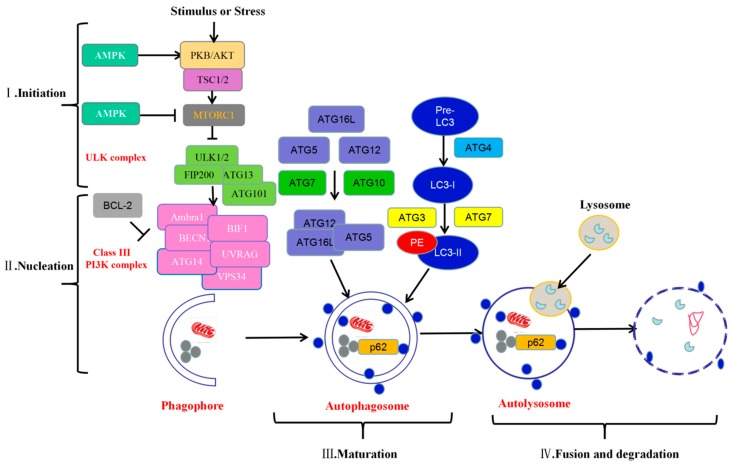
The molecular mechanism of mammalian autophagy regulation. The autophagy process consists of several stages, including initiation (I), nucleation (II), maturation (III), and fusion and degradation (IV). The same color indicates the involvement of a protein or molecule in a complex; blue circles indicate autophagosomes; gray circles indicate lysosomes.

**Figure 2 viruses-11-00776-f002:**
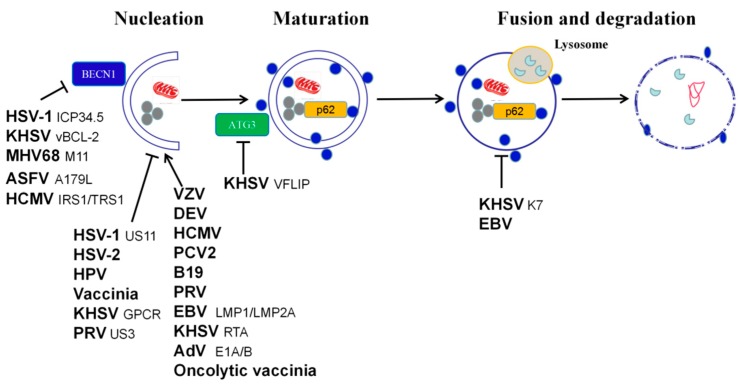
Viral regulation of the autophagy pathway. Several DNA virus-encoded proteins interact with Beclin-1 to inhibit the nucleation of autophagosomes, including herpes simplex virus type 1 (HSV-1) ICP34.5, Kaposi’s sarcoma-associated herpes virus (KSHV) vBCL-2, murine gammaherpesvirus 68 (MHV68) M11, African swine fever virus (ASFV) A179L, and human cytomegalovirus (HCMV) IRS1/TRS1. Other virus-encoded proteins of HSV-1 US11, KHSV GPCR, and pseudorabies virus (PRV) US3, and unknown mechanisms of HCMV, HPV, and vaccinia virus, inhibit autophagosome formation. In contrast, several DNA virus-encoded proteins, such as Epstein–Barr virus (EBV) LMP1/LMP2A, HSV-1 RTA, and ADV E1A/B, and unknown mechanisms of varicella zoster virus (VZV), duck enteritis virus (DEV), HCMV, porcine circovirus type 2 (PCV2), human parvovirus B19 (B19), PRV, and oncolytic adenovirus induce autophagosome formation. KHSV and EBV prevent autophagosomes from fusing with lysosomes to avoid degradation. Arrows show stimulation, whereas other symbols show inhibition.

**Figure 3 viruses-11-00776-f003:**
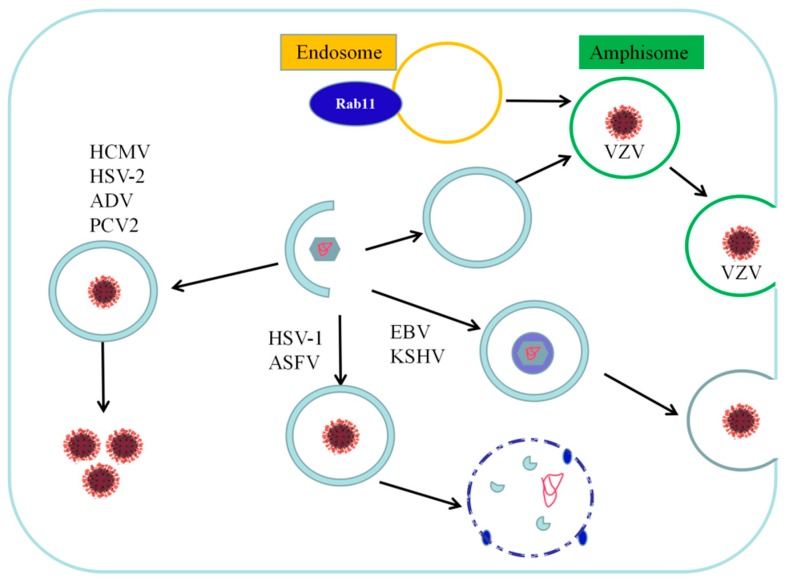
Effects of autophagy on the viral life cycle. After the autophagosome is formed, it fuses with endosomes in the cell to form the intermediate amphisome, which contains only one or several varicella zoster virus (VZV) virions and may be used to release vesicles from the cell. Autophagosomes transport Kaposi’s sarcoma-associated herpes virus (KSHV) and Epstein–Barr virus (EBV) particles to the cell surface. In addition, autophagy promotes viral packaging and assembly, and the autophagy pathway (LC3) is found in viral particles, indicating that EBV subversion of autophagic machinery generates a virion envelope. Autophagy inhibits the replication of several DNA viruses, such as herpes simplex virus type 1 (HSV-1), herpes simplex virus type 2 (HSV-2), and African swine fever virus (ASFV), but promotes the replication of several other DNA viruses, such as duck enteritis virus (DEV), ADV, and porcine circovirus type 2 (PCV2), by influencing the life cycle of the infected host cell; however, the specific mechanisms remain unclear.

**Figure 4 viruses-11-00776-f004:**
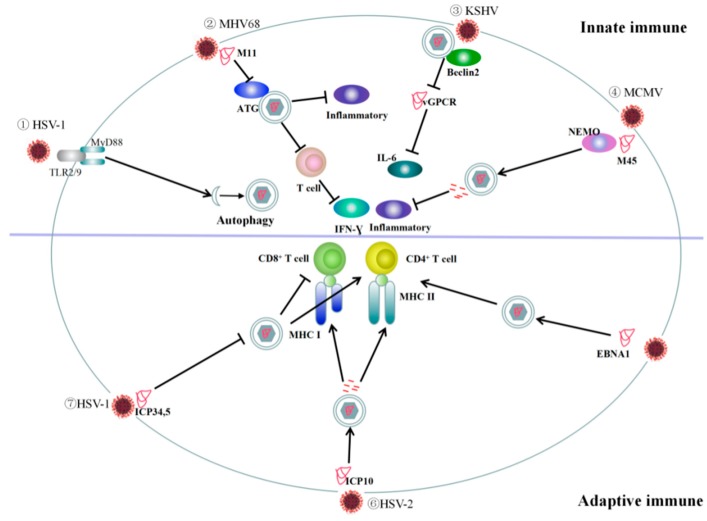
Interactions between autophagy and immunity in response to viral infection. ① Herpes simplex virus type 1 (HSV-1) adsorbs onto the cell surface and triggers the recruitment of the myeloid differentiation factor 88 (MyD88) adapter protein through Toll-like receptor 2 (TLR2) and TLR9, resulting in the activation of autophagy in human acute myeloid leukemia (THP-1) cells. ② M11 binds to autophagy-related genes (ATG) to inhibit autophagy, viral-induced systemic inflammation, and interferon (IFN)-γ production in T cells. ③ It has been demonstrated that lysosomal-dependent Beclin-2 reduces viral G-protein coupled receptor (GPCR) levels and viral GPCR-induced IL-6 signaling. ④ The murine cytomegalovirus (MCMV) protein M45 binds to NF-κB kinase subunit gamma (NEMO/IKKγ) and is delivered to autophagosomes and transported to lysosomes for degradation, resulting in attenuation of the host inflammatory response. ⑤ Plasmacytoid dendritic cells (DCs) infected with Epstein–Barr virus (EBV) release type I IFNs in response to TLR activation and autophagy. ⑥ HSV-2 infectious cell culture protein 10 (ICP10) is delivered to the autophagosome/lysosomal degradation pathway, thereby significantly increasing major histocompatibility complex (MHC) class I and class II antigen presentation. ⑦ HSV-1 ICP34.5 and its Beclin-binding domain suppress DC autophagy and increase MHC class II presentation capacity.⑧ ICP34.5-mediated autophagy inhibits the presentation of endogenous viral antigens to MHC class I molecules in primary DCs.

**Table 1 viruses-11-00776-t001:** Summary of known interactions between human or animal DNA viruses and autophagy.

Family/Virus	Host	Interactions with Autophagy	Impact of Autophagy on Virus Replication	Reference
***Adenoviridae***				
Oncolytic adenovirus	Human	Oncolytic adenoviruses induce autophagy	FADD-induced enhancement of autophagy contributes to viral replication and virus spread	[50,51]
Fowl adenovirus serotype 4 (FAdV-4)		FAdV-4 induces autophagy of hepatocytes		[52]
***Herpesviridae***				
Herpes simplex virus type 1 (HSV-1)	Human	Regulation of autophagy by HSV-1 is cell type-dependent Most studies report detrimental effects of autophagy during HSV-1 infection	Transient activation of autophagy in THP-1 cells via MyD88 adaptor protein is beneficial for viral entry HSV-1 particles can be degraded by autophagy	[53,54,55]
Herpes simplex virus type 2 (HSV-2)	Human	Autophagy seems to be controlled in HSV-2-infected fibroblasts	Basal autophagy promotes viral replication in fibroblasts	[56]
Varicella zoster virus (VZV)	Human	Activates complete autophagy Inhibits autophagic flux	VZV induces complete autophagic flux to help viral propagation VZV titers are higher when autophagic flux is inhibited versus upregulated	[57,58,59]
Duck enteritis virus (DEV)	Waterfowl	Activates complete autophagy	DEV induces complete autophagic flux to help viral propagation	[60]
Pseudorabies virus (PRV)	Pig	Inhibition of autophagy PRV induces autophagy via the classical Beclin-1-ATG7-ATG5 pathway	Autophagy inhibits PRV replication and infection Enhances viral replication in N2a cells in vitro	[61,62]
Human cytomegalovirus (HCMV)	Human	Infection stimulates autophagy and subsequently blocks autophagosome degradation	Autophagy proteins or membranes participate in viral propagation	[63,64]
Murine cytomegalovirus (MCMV)	Mouse	Induces autophagy during early stages of infection and then subsequently blocks it	Blocks the autophagic flux leading to an accumulation of autophagosomes, which helps viral propagation	[65]
Kaposi’s sarcoma-associated herpesvirus (KSHV)	Human	During latency, HHV8 encodes a vFLIP homolog that inhibits autophagy by interacting with ATG3 Autophagy is stimulated during HHV8 reactivation and RTA alone induces autophagosome formation in both 293T and B cells	During latency, autophagy inhibition blocks oncogene-induced senescence Evidence of viral particle transport in autophagosomes and a positive role for autophagy during viral reactivation	[66,67,68]
Epstein–Barr virus (EBV)	Human	During the lytic cycle: autophagic flux is blocked and autophagic vacuoles are hijacked by the virus for envelopment/egress During latency: autophagy stimulation by LMP1 and LMP2A favors cell survival	During the lytic cycle: EBV may limit lysosomal degradation of viral components and hijack the autophagic vesicles for its own benefit During latency: EBV can benefit from autophagy	[69,70]
Rhesus monkey rhadinovirus	Rhesus monkey	During latency, vFLIP-induced autophagy protects cells from apoptosis		[71]
Murid herpesvirus 68	Mouse and small rodents	During latency, MHV68 expresses a viral homolog of Bcl-2 named M11 that blocks autophagy by interaction with Beclin-1	Autophagy allows virus reactivation from latency	[72]
***Poxviridae***				
Vaccinia virus	Human	VV-Onco induces autophagy in MHCC97-H cells	Cellular autophagy machinery is not required for vaccinia virus replication and maturation	[73,74]
***Circoviridae***				
Porcine circovirus	Pig	PCV2 induces autophagy in PK-15 cells	Uses autophagy machinery to enhance its replication in PK-15 cells	[75,76]
***Parvoviridae***				
B19 virus	Human	Mitochondrial autophagy is specifically found in B19-infected cells	Inhibition of autophagy by 3-MA significantly facilitates B19-infection-mediated cell death	[77]
***Papillomaviridae***				
Human papillomavirus (HPV)	Human	Activated mTOR phosphorylation can inactivate ULK1, thereby inhibiting autophagosome formation	HPV inhibits autophagy to promote infectivity	[78]
***Asfarviridae***				
African swine fever virus (ASFV)	Pig	ASFV does not induce autophagy in infected cells	Induction of autophagy reduces the number of infected cells	[79]
***Hepadnaviridae***				
Hepatitis B virus (HBV)	Human	HBV can induce autophagy in vitro and in vivo	HBV proliferation is suppressed upon inhibition of autophagy	[80,81]
***Polyomaviridae***				
JC virus	Human		Autophagy degrades JC viral proteins	[82]
Simian virus 40 (SV40)	Simians	SV40 ST antigen activates AMPK, inhibits mTOR, and induces autophagy		[83]
BK polyomavirus (BKPyV)	Human		Autophagy promotes BKPyV infection	[84]
***Nimaviridae***				
White spot syndrome virus (WSSV)	Shrimp	During early stages of viral infection, shrimp autophagy is induced	Host autophagy facilitates viral infection in vivo	[85]
***Baculoviridae***				
*Bombyx mori* nuclear polyhedrosis virus (BmNPV)	Silkworm	BmNPV infection can trigger autophagy	The virus may utilize the host autophagy mechanism to promote its own infection process	[86]
***Iridoviridae***				
Infectious spleen and kidney necrosis virus (ISKNV)	Fish	ISKNV induces autophagy of cells during the early stages of infection		[87]
Iridovirus	Fish	Autophagy is induced during early infection of primary renal cells in Chinese giant salamander		[88]

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
