# Peer review of "Interactions between Autophagy and DNA Viruses"

_viruses, 2019, doi:10.3390/v11090776_

Round 1
Reviewer 1 Report
This is a comprehensive review that summarizes a good amount of information about autophagy in DNA viruses. Perhaps some minor updates could be considered since some recent publications particularly in HSV field have implicated additional viral genes and additional roles of autophagy during infection.
Author Response
Major issues:
Point 1 Introduction of autophagy. There are many misleading points. p62 is one of an increasing pool of selective autophagy receptors. The authors should mention the others, or readers will misunderstand that p62 is the only one that links ubiquitinated substrates to lysosomes. ….
Response 1:Others selective autophagy receptors have been mentioned,please see lines 44-48.
Point 2 induce activation of autophagy through induction of mTOR…. activates mTOR complex 1 (mTORC1), subsequently inhibiting autophagy. Does induction of mTOR induce activation of autophagy? one of these statements is not correct. …..mTORC1 activates the unc-51-like autophagy activating kinase (ULK)1/2 complex…. Activates?There are some mTOR/mTORC1-independent autophagy mechanisms.
Response 2:The wrong description has been corrected,please see lines 61 and 68;There are some mTOR/mTORC1-independent autophagy mechanisms also have been mentioned,please see lines 69-71.
Point 3 Section 1.2. Multiple roles of autophagy. The subtitle should be changed to “Multiple roles of autophagy in immune responses” in that autophagy has many other roles.
Response 3: The subtitle has be changed to “Multiple roles of autophagy in immune responses”,please see line 106.
Point 4 Section 1.2.1. Autophagy in innate immunity. The breakthrough research advances on the interplay between autophagy and cGAS-STING-mediated innate immunity is not included.
Response 4:The breakthrough research advances on the interplay between autophagy and cGAS-STING-mediated innate immunity has been included,please see lines133-137.
Point 5 Important recent related publications are missing. For example, for the interaction between EBV and autophagy, a recent paper shows that p62-mediated selective autophagy is induced by EBV latent infection.
Response 5:I have re-searched for all the latest research progress, some missing important recent related publications have been added.Please see lines 216-217,238-241,290-292,402-405,460-462,550-553.
Minor:
Point 1“or” should be “and”. Table 1.
Response1:“or” have been replaced with “and”,please see line 27.
Point 2 Lines should be added to separate the contents for each virus. The current style is very confusing, and readers cannot tell which virus a given description in the right columns is talking about.
Response2: Table 1. Lines have been added to separate the contents for each virus.Please see the Table1.
Point 3 Line 27. Line 373. Cited a wrong reference [79].
Response3: The cited a wrong reference have been corrected.Please see line 379.
Reviewer 2 Report
RE: viruses-575462-revision
This review focuses on the interaction of DNA viruses and autophagy. It is well organized, with three parts: (1) Introduction of autophagy and its roles in immunity; (2) Regulation of autophagy by DNA viruses, and (3) The roles of autophagy in DNA virus infection.
There are some issues, including that recent breakthrough advances in this field is not mentioned, which should be addressed before acceptance.
Major issues:
Introduction of autophagy. There are many misleading points. p62 is one of an increasing pool of selective autophagy receptors. The authors should mention the others, or readers will misunderstand that p62 is the only one that links ubiquitinated substrates to lysosomes. …. induce activation of autophagy through induction of mTOR…. activates mTOR complex 1 (mTORC1), subsequently inhibiting autophagy. Does induction of mTOR induce activation of autophagy? one of these statements is not correct. …..mTORC1 activates the unc-51-like autophagy activating kinase (ULK)1/2 complex…. Activates?? There are some mTOR/mTORC1-independent autophagy mechanisms. Section 1.2. Multiple roles of autophagy. The subtitle should be changed to “Multiple roles of autophagy in immune responses” in that autophagy has many other roles. Section 1.2.1. Autophagy in innate immunity. The breakthrough research advances on the interplay between autophagy and cGAS-STING-mediated innate immunity is not included. Important recent related publications are missing. For example, for the interaction between EBV and autophagy, a recent paper shows that p62-mediated selective autophagy is induced by EBV latent infection.
Minor:
Line 27. “or” should be “and”. Table 1. Lines should be added to separate the contents for each virus. The current style is very confusing, and readers cannot tell which virus a given description in the right columns is talking about. Line 373. Cited a wrong reference [79].Author Response
Point1 This is a comprehensive review that summarizes a good amount of information about autophagy in DNA viruses. Perhaps some minor updates could be considered since some recent publications particularly in HSV field have implicated additional viral genes and additional roles of autophagy during infection.
Response1:Updates publications have been added.Please see lines 216-217 and 238-241.